# Recent Progress and Challenges Regarding Carbon Nanotube On-Chip Interconnects

**DOI:** 10.3390/mi13071148

**Published:** 2022-07-20

**Authors:** Baohui Xu, Rongmei Chen, Jiuren Zhou, Jie Liang

**Affiliations:** 1School of Microelectronics, Shanghai University, Shanghai 201800, China; xbh1@shu.edu.cn; 2Interuniversity Microelectronics Centre (IMEC), 3001 Leuven, Belgium; rongmei.chen@imec.be; 3Emerging Device and Chip Laboratory, Hangzhou Institute of Technology, Xidian University, Hangzhou 311200, China; zhoujiuren@163.com

**Keywords:** on-chip interconnect, carbon nanotube, through-silicon-via (TSV), Cu-CNT composite

## Abstract

Along with deep scaling transistors and complex electronics information exchange networks, very-large-scale-integrated (VLSI) circuits require high performance and ultra-low power consumption. In order to meet the demand of data-abundant workloads and their energy efficiency, improving only the transistor performance would not be sufficient. Super high-speed microprocessors are useless if the capacity of the data lines is not increased accordingly. Meanwhile, traditional on-chip copper interconnects reach their physical limitation of resistivity and reliability and may no longer be able to keep pace with a processor’s data throughput. As one of the potential alternatives, carbon nanotubes (CNTs) have attracted important attention to become the future emerging on-chip interconnects with possible explorations of new development directions. In this paper, we focus on the electrical, thermal, and process compatibility issues of current on-chip interconnects. We review the advantages, recent developments, and dilemmas of CNT-based interconnects from the perspective of different interconnect lengths and through-silicon-via (TSV) applications.

## 1. Introduction

Transistors and interconnects are vital components of integrated circuits (IC). With the advancement of technology nodes and the reduction of critical dimensions, the performance of transistors has been greatly improved, while the interconnect has become the bottleneck limiting the development of IC.

Since the 1990s, transistors are no longer the only major factor affecting integrated circuits, and interconnects have gradually become another breakthrough for improving chip performance. With scaling down, in the technology node of 180 nm, the demands of high performance and reliability drove the transition from Aluminum (Al) interconnects to Dual Damascene Copper (Cu) interconnects due to the better conductivity and electromigration (EM) lifetime of copper [1]. TEM images of the EM effect are shown in Figure 1. A few years later, in 2003, low-K dielectric was introduced to separate copper lines, reducing parasitic capacitance, enabling faster switching speeds, and lowering heat dissipation [2]. Low-K dielectric is one of the strategies employed to carry out scaling followed by Moore’s law. Since then, the composition of Cu and low-K material has become the basic structure of integrated circuits. Figure 2a show a simple conventional interconnect structure.

The main concerns with the continued scaling of copper-based interconnects are the significant increase in resistivity and electromigration issues resulting from the surface, grain boundary, and line edge roughness scattering [3]. In the back-end-of-line (BEOL) integration of copper-based interconnects, copper is surrounded by a barrier and liner. A barrier is necessary to maintain a certain thickness of about 2 nm to prevent the diffusion of copper through the interlayer dielectric (ILD); therefore, the effective conducting part of copper decreases with continuously scaling [4]. The decreasing dimension leads to higher resistivity due to electron scattering, as shown in Figure 2b, and reduced electromigration resistance. Copper-based interconnects are now facing similar challenges to Aluminum-based interconnects in terms of performance and reliability. In this case, recent research proposed by the IMEC (Interuniversity Microelectronics Centre) suggested the replacement of copper with Ruthenium (Ru) and Cobalt (Co) [3,5]. Although the resistivity of Ru and Co is higher than the resistivity of Copper, the interconnect lines developed with Ru and Co can conduct without barrier layers, which shows better overall conductance than that of copper lines. Furthermore, the fabrication of interconnect lines using Ru and Co is compatible with the process used currently in the back end of the line, which means lower replacement costs.

Recently, in advanced technology nodes, the semiconductor industry has focused on seeking a candidate to replace copper. We summarize the recent development of on-chip interconnect alternatives in Table 1. In the 14 nm node, Intel studied Copper and Tungsten (W) for new processes and new barrier materials [8,9]. In 10 nm and 7 nm nodes, Intel and Global Foundries explored the use of Co without barriers [5,10]. Regarding 5 nm and 3 nm nodes, IMEC expanded the possibility of employing Ru, as its resistivity is dependent on thickness [5,10,11,12]. When going beyond 3 nm nodes, it is necessary to find new materials, such as carbon-based materials, to meet the urgent demands of downscaling for interconnects in back-end-of-line fabrication. TSMC (Taiwan Manufacturing Company) has shown interest in multilayer graphene nanoribbon (GNR), which exhibits superior performance in intermediate and global interconnects [13,14]. Carbon nanotubes (CNTs) are the materials applied in interconnects with great potential due to their long mean free paths (MFP) [15], high thermal conductivity [16], large current carrying capacity [17], and excellent mechanical properties [18]. However, contact resistance and integration processes for CNT-based interconnects remain a great challenge for the application of CNT-based on-chip interconnects [19].

In this paper, we mainly discuss recent developments and challenges in CNT-based interconnects. For local interconnects, size effects are the main issue due to the shrinking of critical dimensions. We discuss the conventional local interconnect physical resistance limitations and capacitance drawbacks and how single-walled (SWCNTs) and multi-walled CNTs (MWCNTs) could potentially solve these challenges by means of doping. Non-ideal factors in the practical CNT integration process are also covered. At the intermediate level of interconnects, performance is dominated by the product of resistance and capacitance due to larger transmission lines. Additionally, the ballistic transport properties of CNTs are highlighted gradually in intermediate levels. We analyze CNT-based interconnects with regard to their electrical properties. Among them, double-walled carbon nanotube (DWCNT) shows its particularity but still requires the support of specific processes. In the global interconnect, the reliability of the transmission line is especially important. We not only discuss the electrical and thermal properties of CNT-based interconnects but also give more attention to the analysis of reliability. The significant advantages of CNT-based interconnects in global interconnects are revealed. We introduce the process progress and high-frequency characteristics of CNT TSVs, and the feasibility of Cu-CNT TSVs is overviewed. Fault diagnosis techniques and the CNT-immune technique are also considered.

A summarized timetable is shown in Figure 3, which includes the main research on the electrical, thermal, and power analysis of CNT-based interconnects in recent decades. We notice that the electrical properties of CNT-based interconnects are the main point of interest, and there is a higher propensity for CNT-based interconnects to be applied on the global stage. Following CNT interconnect technology evolutions, the first compact model of standalone CNT was established in 2002 [20]. Based on that, complex electrical models considering more physical properties on CNT bundles and MWCNTs for interconnect applications were developed further in 2005 [21] and 2008 [22]. Regarding thermal analysis, one of the most important studies, conducted in 2005 [23], used three-dimensional finite element electrothermal simulations to conduct a comprehensive analysis of the thermal power of CNT-based interconnects. Based on this method, in 2009, the characteristics and requirements of the SWCNT bundle used as VLSI interconnects were discussed in depth [24]. Moreover, the first CNT interconnect integration process was introduced by Franz Kreupl in the year 2002 [25]. Decades after that, various integration processes were developed [26], and the most Silicon-compatible CVD (chemical vapor deposition) growth process was achieved at around 550 °C. TSVs using CNTs as materials were first discussed in 2007 [27]. The first generation mentioned CNT bundles can be uniformly grown to bond two wafers, namely through-wafer-interconnects at that time. In the following years, the technology of CNT TSV was continuously developed and advanced. In 2016, a unique process for fabricating Cu-CNT composite TSVs was proposed [28]. It demonstrated the advanced TSV process and included thermal analysis, which provided experimental foundations for subsequent development. Additionally, the Cu-CNT composite used for interconnect applications was revealed to be the potential solution for global interconnects in 2013 [29]; its electrical modeling investigations are presented in [30,31].

The following content will be divided into four parts. Section 2 analyzes the advantages and disadvantages of CNTs in view of physical models according to different interconnect lengths (local, intermediate, and global). Section 3 presents the application of CNT and Cu-CNT composites on through-silicon-via (TSV). The discussion and perspectives will be driven in Section 4. Conclusions will be presented in Section 5.

## 2. On-Chip Interconnect

Interconnects can be classified as local, intermediate, and global, depending on their typical dimensions. We summarized the typical length values for the three different interconnect stages in Table 2 for easy distinction. However, it should be emphasized that the three interconnect stages are not divided by precise values but by a concept of scope. Local interconnects are the shortest and the narrowest, ranging from several nanometers to a few micrometers. They connect various transistors and logic blocks. The length of intermediate interconnects is located between the local and global interconnects, ranging from a few micrometers to tens of micrometers. Intermediate interconnects connect complex logic blocks and different cores. Global interconnects have a length greater than hundreds of microns. Global interconnects usually transmit power, ground, and clock signals to all the cores in circuits.

### 2.1. Local Interconnect

Local interconnects are on the lowest levels of the interconnect stack connecting nearby transistors and logic blocks, usually at the nanoscale. Logically, they are susceptible to the size effect, which leads to circuit performance degradation and tends to worsen with future device miniaturization. As shown in Figure 4, narrower wires have smaller grains and larger grain boundary densities, which increase electron scattering and thus increase resistance. Moreover, local interconnects occupy over 50% of power dissipation among interconnects [59]. The performance of local interconnect lines is vital to the chip.

In addition to physical resistance limitation, capacitance is also a critical issue that affects local interconnects. For nanoscale interconnect lines, their resistances are too small to be comparable to those of transistors and blocks, while their capacitances almost remain unchanged except for the effect of crosstalk [39]. The capacitance, in that case, which depends on the geometry and the dielectric constant of the surrounding insulating material, mainly contributes to the performance of transmission lines over short distances by affecting RC delay. Furthermore, downscaling leads to a higher area density, indicating that the distance between the adjacent wires has become smaller. The three primary capacitance components (the line-to-line capacitance, the line-to-ground capacitance, and the cross-over capacitance), as indicated in Figure 5, are all inversely proportional to the spacing between the wires [61]. Meanwhile, crosstalk, another critical issue, arises. In the empirical model and aggressor-victim model [61,62], the width, thickness, height, spacing between neighboring wires, the aspect ratio of wires, and switching signals are all influencing parameters of the crosstalk effect. In particular, the distance between adjacent wires decreases, leading to higher coupling capacitance. Coupling capacitance can delay the signal propagation in wires and result in logic errors and noises, so-called crosstalk-induced delay [62,63]. With further scaling, the negative impacts caused by capacitance and crosstalk greatly increase.

Carbon nanotubes have shown great potential to solve these issues. According to the transmission line model, the capacitance of the SWCNT is composed of the quantum capacitance and the electrostatic capacitance in series, which is theoretically smaller than that of traditional Cu wires [20]. Compared to copper-based interconnects in 22 nm nodes, for local interconnects, several layers of SWCNT can offer up to 50% reduction in capacitance and power dissipation with up to 20% in latency if they are short enough (<20 µm) [39,64]. For the MWCNT, the shell-to-shell capacitance needs to be further considered. However, it is more complicated and needs to be analyzed according to the actual application. MWCNTs exhibit up to 10% improvement in capacitance for local interconnects compared to copper-based interconnects in 14 nm nodes [22]. Logically, reduced capacitance leads to better performance on crosstalk. Research shows that crosstalk-induced overshoot/undershoot remains nearly the same as technology node scaling and increases as the length of the copper-based interconnect increases. Whereas in CNT-based interconnects, the overshoot/undershoot remains the same and is almost unaffected by the scaling of interconnect length [43]. However, CNTs do not perform as well as expected in local interconnects. In short-distance interconnects, although the delay is mainly capacitance-dominated, fabrication process factors such as contact resistance, CNT chirality, and CNT density are also taken into account. These non-ideal factors are unavoidable in practical processes and degrade the electrical properties of CNTs. Especially in the local stage, CNTs cannot entirely take advantage of their high electron mobility, which makes the performance of CNT-based interconnects far worse than ideal. In addition, the advantage of MFP length cannot be fully benefited, and the advantage of capacitance is not significant. The actual delay of CNT is usually greater than that of Cu [22]. Theoretically, in the case of short local interconnects, CNT-based interconnects require harsh conditions to potentially outperform copper-based interconnects, such as in all-carbon circuits, and warrant a fairly small CNT pitch in CNTFETs (carbon nanotube field-effect transistors) (<9 nm) [65].

Generally, at nanoscale interconnect lengths, compared to Cu lines, SWCNT bundles and MWCNT have better performance in terms of delay, as well as multilayer SWCNT interconnects [22,24,39]. Recently, Chen et al. studied the impact of charge transfer doping on the performance and variability of MWCNT interconnects using enhanced compact models [53,54]. They further explored an all-carbon SRAM (ACS) (SRAM: Static Random-Access Memory) using a 5 nm technology node, which showed a great improvement in power efficiency with 72% lower energy-delay-product (EDP) and 48% lower static power, despite a slight speed reduction, compared to 7 nm FinFET (fin field-effect transistor) SRAM cells with copper-based interconnects [56,66].

From the point of view of CNT-based interconnect integration, both the capacitance and delay of CNT-based interconnects still have room to be improved compared to copper-based interconnects in the local interconnect region due to large CNT-metal contact resistance, the density of CNT bundles, and so on [39,43,44,67]. However, the study in [33] showed that MWCNTs can exhibit conductivity surpassing that of Cu lines in local interconnects by doping with certain concentrations of Pt salts [55,68].

For local interconnects, doping of CNT provides a potential solution to improve the problems caused by increased capacitance and gradually deteriorated crosstalk and has an advantage regarding power efficiency; however, CNT-based interconnects still require further study to overcome CNT–metal interface contact resistance with respect to the current transistor fabrication process in order to outperform Cu line local interconnects [22].

### 2.2. Intermediate Interconnect

Intermediate interconnects are usually at a micron-meter scale. They mainly serve to connect logic blocks and cores. At this level, interconnect delay depends mostly on the product of resistance and capacitance due to larger transmission lines. Along with advanced technology nodes, the shrinking of Cu lines leads to the reduction of effective conducting parts of interconnect lines, which further impact the performance of intermediate interconnects. Moreover, repeaters are used at the intermediate level to increase the drive capability and reduce the signal delay, but bring some negative effects regarding power dissipation, area resources, and design [69,70].

The long MFP and ballistic transport properties of CNTs are highlighted gradually in intermediate interconnects [15,71,72]. For densely packed bundles at the intermediate interconnect level, the bundle resistance is less than that of Cu for a wide range of interconnect lengths, even for low metallic CNT density (45%) [24]. Dense SWCNT bundle interconnects can easily surpass Cu with at least 30% less latency [24]. Similarly, the resistivity of MWCNTs could be several times lower than that of Cu wire and becomes increasingly comparable to that of SWCNT bundles for long lengths (>10 µm) [22,38]. CNTs not only improve latency but also advances the use of repeaters. With the same repeater size, carbon-based interconnects have significantly less delay than that of copper-based interconnects. The optimal number of repeaters could also be reduced [41,47].

Additionally, research shows that both DWCNT and SWCNT bundles provide a significant improvement in performance compared to Cu wires [40]. In particular, the performance of DWCNT is even better than that of SWCNT. In the 14 nm technology node, DWCNT has a 20% improvement in crosstalk-induced time delay over SWCNT at the intermediate level [42]. From a process point of view, DWCNT is easier to achieve with a high metallic chirality ratio than SWCNT, and it also has small capacitance [57]. Therefore, in the intermediate stage, high-density DWCNT bundles could be a better solution under the consideration of delay based on RC product, as mentioned above. However, in general, DWCNT is far from SWCNT and MWCNT in terms of application range and process compatibility, and there are few studies investigating the feasibility of DWCNT. Therefore, the potential application prospects of DWCNT and the targeted process issues remain to be further explored.

### 2.3. Global Interconnect

Global interconnects are usually more than 100 micronmeters. They carry power, ground, and clock signals, which indicate a high current carrying capacity. Therefore, for global interconnects, performance improvement should be considered, but more importantly, the optimization of interconnect reliability should be taken into account to meet the requirements in advanced nodes.

At the global level, a reduction in the size of Cu lines results in the increase of resistance and current density, which in turn causes critical issues on delay, IR drop, power dissipation, and electromigration effect. Moreover, the number of repeaters grows with interconnect length, increasing power consumption while reducing latency. Under certain conditions, it could cause a blockage problem, which reduces the transmission efficiency [73]. Power consumption is not only a problem for energy transmission efficiency but also a thermal problem caused by high current flow. Thermal management has naturally become one of the key optimization directions.

In the global interconnect with resistance as the main factor affecting delay, CNT takes full advantage of its high mobility and long MFP, surpassing Cu with lower resistance. For the global interconnect scale, CNT-based interconnect lines have at least 20% lower resistance compared to that of Cu lines and an optimized RC delay of more than 30%, which greatly enhances the performance of interconnecting transmission [22,39,74,75]. Additionally, because the transmission length is long enough, the negative effects of non-ideal factors such as contact and defects are relatively reduced [22,50,76]. Theoretically, the transport speed of CNT for long-distance transport is much better than that of Cu. It is worth mentioning that some studies have shown the mixed-CNT bundle has better performance than using SWCNT bundles [48,77]. However, it is limited by complex distribution requirements and process challenges; its practical application still needs further investigation. Figure 6a–g show the SEM images of CNT vias. They display the different steps in the CNT growth process. They demonstrate that the width, length, and area density of CNTs in the growth process are difficult to precisely control. Each step in the process causes changes in the CNTs, as the CNTs are highly variable. The idea of mixed CNT bundles is practical for interconnect processes. However, using the Gaussian distribution to simulate the change of CNT diameters at different positions still has a certain discrepancy with the actual situation, so further exploration is required. The challenge of dimension control is also applicable to SWCNT bundles and MWCNTs.

A densely packed SWCNT bundle can reduce power consumption by up to eight times compared to Cu-based interconnects at the 14 nm node [24]. Theoretically, at a length of 100 microns, its EDP can be 12 times better than Cu, reducing the energy-per-bit by at least three times at the 11 nm node [45]. The optimization of power consumption is mainly attributed to the low resistance, low capacitance, and excellent thermal properties of CNTs. Power consumption issues and thermal issues are often side-by-side.

According to molecular dynamics simulations, the thermal conductivity of CNTs can reach 6600 Wm^−1^K^−1^ [78]. It depends on multiple factors such as CNT length, diameter, defects, bundle density, etc. [79]. Therefore, CNTs, under different conditions, may be excellent thermal conductors or thermal insulators [80]. Experiments show that the thermal conductivity of SWCNTs with a length of 2.6 microns and a diameter of 1.7 nm can reach 3500 Wm^−1^K^−1^ at room temperature, while the thermal conductivity of a single MWCNT with a diameter of 14 nm is more than 3000 Wm^−1^K^−1^ at room temperature [52,81]. In contrast, the thermal conductivity of copper is only about 400 Wm^−1^K^−1^. Moreover, CNTs can maintain the basic stability of the structure at a temperature of 1000 K, which is enough to cover the upper temperature limit of on-chip interconnects at present [46,82]. CNTs are excellent thermal conductive materials that can be used to enhance interconnect reliability. Table 3 summarize the electrical conductivity, thermal conductivity, electron mean free path, and dielectric constant of various interconnect-based materials.

In CNT thermal models for circuit-level simulations, the temperature is usually introduced as an inverse function of MFP to show its effect on electron–phonon scattering [90,91]. When the interconnect length is longer than the MFP of CNT, the electron–phonon scattering in the length direction of CNT will cause self-heating and temperature increase. Increased temperature, in turn, enhances scattering and degrades the performance. On the other hand, for large-diameter CNTs, the band gap between sub-bands is smaller than that for small-diameter CNTs. The rise of temperature provides sufficient thermal energy for the carriers to cross the bandgap. That is, the increase in temperature can also increase the number of conducting channels and enhance the transport ability of CNTs [49]. According to the model results, at the global length, the temperature is proportional to the resistance of the CNT [92]. The TCR (temperature coefficient of resistance) value of MWCNT with a diameter of 50 nm at 100 µm length is 2.7, while the TCR value of Cu is 4 [49]. This behavior can be explained as follows: for large-diameter MWCNTs, the increase in the number of conducting channels caused by the temperature increase reduces the scattering resistance and contact resistance so that the resistance growth and self-heating effects are not significant compared to Cu. Therefore, CNTs have less resistance sensitivity to temperature. Combined with their lower resistance and inherently better thermal conductivity, CNT-based interconnects have lower power dissipation and better heat dissipation. For ideal SWCNT bundle interconnects at dimensions of 10–20 nm wide and 1 mm long, the delay is five times less than that of Cu interconnects under the same conditions over the temperature range from 300 K to 373 K [90]. As shown in the simulation in Figure 7, the maximum temperature of CNT-based via is almost two times better than that of Cu via [23,24]. In Figure 7, it can also be clearly observed that CNTs can reduce the maximum temperature regardless of the length of the interconnect used. It demonstrates that the heat dissipation capability of CNTs is superior to that of Cu.

The electromigration problem is especially significant in global interconnects, where interconnect lines carry large current flows that may lead to voids or even circuit failure, as shown previously in Figure 1. Electromigration is a mass transport process due to the self-diffusion of metallic ions in response to an electric field applied across interconnects [93]. With the scaling down of dimensions, Cu lines shorten the EM failure time because the maximum allowed current density decreases with dimension. Research shows that, for Cu interconnects, the normalized median EM failure time approximately degrades 100% scaling from the 180 nm node to the 14 nm node [94]. In contrast, CNT-based interconnect lines have a large current carrying capacity of 109 A/cm^2^, at least two orders of magnitude higher than the maximum current density of Cu interconnect [95]. This shows that using CNT-based interconnects could be a potential solution to alleviate EM problems. In addition to electromigration, time-dependent-dielectric-breakdown (TDDB) is also an important reliability issue, which limits the spacing between adjacent wires for a specific low-k dielectric. After long-term use, the drift of copper ions can cause the deterioration and breakdown of the interconnect structure [96]. The TDDB-induced damage is shown in Figure 8. It indicates that after the TDDB test, the TaN/Ta barrier was damaged at the bottom corner, and more Cu atoms migrated into SiO_2_ as the test time increased. This phenomenon seriously affects the performance and reliability of the interconnect. Therefore, both TDDB and EM are regarded as the key issues surrounding interconnect reliability. CNT-based interconnects are expected to use dielectrics with a lower dielectric constant (*k* < 2.5) [89], thus mitigating the TDDB problem.

## 3. Through-Silicon-Via (TSV)

Through-Silicon-Via is a vital part of modern three-dimensional (3D) integration technology. In the case of high-density integration in the horizontal direction of a single chip, the integration direction is further expanded by stacking multiple layers vertically through TSVs. TSVs further expand the feasibility of continued size reduction and also shorten the interconnect path. TSV usually transmits periodic power and ground signals. It has been widely used in 3D integrated circuits and 3D packaging [98], which has great application prospects in the development of SoC (System on Chip) and heterogeneous integration.

### 3.1. Carbon Nanotube

A commonly used CNT line integration is vertical growth at high temperatures (>700 °C) by catalyst-enhanced chemical vapor deposition (CCVD). There are fewer cases of CNT growth directly in the horizontal direction [57]. Therefore, compared to the horizontal CNT interconnect line process, growing CNTs directly in Vias or TSVs could be a more friendly process and takes full advantage of vertical CNT bundles [9,27]. Using a flip and roll technique to form vertical–horizontal combined structures was investigated in [99].

Contact and CMOS process compatibility are the two major challenges of CNT growth, which affect the resistance and reliability of CNT-based interconnects. Studies have shown that aligned CNTs can be grown vertically as TSVs by different methods at a temperature below 550 °C that is compatible with the CMOS process and device fabrication [34,35]. Experiments show that the aspect ratio (AR) of CNT TSV with a length of 50 µm can achieve 5 or 10, while the resistance is only 69.7 Ω [33]. Moreover, CNTs have excellent thermal conductivity and thermal stability and are very suitable for long-distance transmission within large current-carrying TSV structures. Recently, a method to fabricate CNT-filled TSV using the vacuum-assisted spin coating of polyimide (PI) liners was reported in [37]. It is possible to manufacture a CNT TSV with a diameter of 15 microns and a depth of 200 microns (AR~13.3) at low cost and low temperature (<~240 °C). The temperature requirements for CNT TSVs growth have been successfully reduced to be compatible with CMOS processes. However, using CNTs as Via or in TSV still remains a challenge for mass production and industrial compatibility [100,101].

For high-frequency applications, with current studies, all metallic SWCNT bundles have lower resistance than Cu due to their long MFP, while MWCNTs have poor high-frequency performance due to their large inductance [32,57]. Moreover, the electrical properties of CNT TSVs are easily affected by changes in kinetic inductance. At high frequencies, the electrical properties of CNTs degrade with increasing kinetic inductance [31].

Therefore, CNT TSVs present great improvement in reliability at the expense of the loss of conductivity, showing excellent stability and scalability. However, although the process compatibility of CNTs used as TSVs has been investigated, more attention is required regarding research on interface contact resistance.

### 3.2. Cu-CNT Composite

Both TSVs and global interconnects need to transmit large currents as well as AC (alternative current) signals, which are prone to EM. Faced with this problem, on the basis of CNT-based interconnects, people thought to combine Cu and CNTs to meet the demand. Cu-CNT composite has been experimentally proved to be a material with high conductance (2.3–4.7 × 105 Scm^−1^) and high current capacity (6 × 108 Acm^−2^) [29,102].

In terms of electrical properties, it makes up for the low electrical conductivity of CNT TSV, reaching a level comparable to that of Cu (5.8 × 105 Scm^−1^). At the same time, the ampacity is almost 100 times that of copper, which is more resistant to the EM effect [103]. The performance at high frequencies remains similar to Cu [31]. Compared with CNT, the Cu-CNT composite has better electrical conductivity and the ability to suppress kinetic inductance changes [104]. In addition, its stability and reliability are much better than Cu, which makes the Cu-CNT composite TSV a potential candidate for high-frequency 3D-IC applications. Moreover, studies have shown that surface modification based on Cu-CNT composite can continue to improve its electrical and mechanical properties [105,106]. Although the improvement effect of surface modification is not significant, it provides a direction for further development.

In terms of thermal properties, Cu-CNT composite has higher thermal conductivity and lower CTE (coefficient of thermal expansion) than copper. When the CNT concentration reaches 250 mg/L, the thermal conductivity of the Cu-CNT composite can reach 637 Wm^−1^K^−1^ [86], which is inferior to CNT (>2000 Wm^−1^K^−1^) but still better than Cu (385 Wm^−1^K^−1^). Compared with pure copper, the Cu-CNT composite with 50% copper can reduce the CTE by 90% [51]. At the circuit level, Cu-CNT TSV shows its excellent thermal properties. For Cu-CNT TSVs, the filling rate of CNTs is inversely proportional to the temperature increment and the maximum temperature [107]. Additionally, it exhibits less temperature sensitivity than Cu in both conductivity and high-frequency transmission performance [108].

From a process point of view, the use of Cu-CNT composite materials can better combine with the existing Cu process and have good interface compatibility. It is possible to achieve void-free filling, CMP (chemical mechanical polishing), or patterning integration and decrease variability [36]. Figure 9 show the fabrication process of Cu-CNT TSVs. Grid arrays of carbon nanotubes are grown by CVD (chemical vapor deposition). After a series of steps, including sputtering, transfer, etc., Cu is injected into the pre-reserved gaps during the electroplating process to realize the Cu-CNT composite, as shown in Figure 10. Unfortunately, the major challenge is the process still needs to be improved and explored to be suitable for VLSI manufacture [109,110]. Specific developments are required for Cu-CNT composites.

In conclusion, although the manufacturing process has yet to be further developed toward industrial requirements, the use of Cu-CNT composite as a TSV or global interconnect material enhances both the performance and reliability of the integrated circuit, even better than CNT in the aspect of electrical performance. In addition, surface modification provides room for further development. Therefore, Cu-CNT composite material is one of the potential solutions for future global interconnects.

## 4. Discussion and Perspectives

With the continuous reduction of the overall size of integrated circuits, entering the nanometer and angstrom era, Cu/low-k interconnects have not been replaced as predicted by some articles [13,111] and have stood the test of time. For CNT-based interconnects, there have been many achievements in physical models, designs, and processes in the past few decades. Most of them indicate that CNT-based interconnects have three advantages, high performance, improved reliability, and better energy efficiency [84], which are suitable for on-chip interconnect applications. Moreover, many studies have shown that there is great potential for CNT-based interconnects to surpass Cu lines under certain conditions.

As far as physical models are concerned, efforts should be made to develop realistic, high-reliability physical models. It is not only necessary to consider the ideal situation but also to further consider various factors such as variability and defect-containing factors. Hierarchical physical models should be considered from material to device to circuit, considering the possible non-ideal effects in the actual fabrication process as comprehensively as possible. In that case, design technology co-optimization (DTCO) should be investigated to provide a theoretical basis and guidance for future CNT-based interconnect architectures. In terms of the CNTs integration process, it should focus on optimizing issues including contact, CNT density and distribution, chirality control, growth temperature, and cost to meet the requirements for VLSI circuit manufacture. In addition, doping of CNT and Cu-CNT composite material needs specific research to enrich the full schema of carbon-based BEOL integration.

Moreover, for CNT-based interconnect circuit-level simulations and behavior analysis, fault diagnosis should be properly incorporated. The purpose is to detect, identify, and isolate various anomalies and faults as early as possible in order to minimize their damage and facilitate the VLSI manufacturing process. However, there is little research in this area. Most of these studies focus on diagnosing and measuring delay faults. A ring oscillator-based testing technique [112] and a model with an inverter chain followed by a D flip-flop [113] are, respectively, proposed to diagnose and measure delay–fault caused by the variability of CNTs [53]. Such techniques help to judge whether the dimension of CNTs is as expected. However, for fault-tolerant technology, the traditional method is not applicable due to the large power consumption and area [84]. Hence, a defective CNT-immune design technique is proposed with a smaller power, area, and speed impact than traditional defect and fault tolerance techniques [114]. It solves the problems of misposition and misalignment of CNTs and has good compatibility with the current VLSI circuit design flows. Nevertheless, the above-mentioned techniques do not comprehensively cover the possible factors of CNT-based interconnect failure, such as the chirality of CNTs where all-metallic CNTs are favorable for interconnect applications. Therefore, the fault diagnosis and tolerance technology for CNT-based interconnects still need to be further explored to serve VLSI circuit manufacturing.

While continuing to delve into the physical nature of CNTs, novel application-oriented optimization designs should also be carried out based on the advantages of CNTs themselves. For example, the application of CNT in N3XT and N3XT 3D MOSAIC structures (N3XT: Nano-Engineered Computing Systems Technology; MOSAIC: MOnolithic/Stacked/Assembled IC) [115,116]. The N3XT structure contains energy-efficient field-effect transistors (FETs), high-density nonvolatile memories, fine-grained monolithic 3D integration, efficient heat removal, and computation immersed in memory. Moreover, CNTs hold the promise of optimizing thermal management in the N3XT 3D MOSAIC.

## 5. Conclusions

In this paper, we reviewed the state-of-the-art CNT-based interconnects from the perspective of different interconnect levels and TSV applications. Carbon nanotubes exhibit different properties at different length stages. Theoretically, although CNTs have a good crosstalk suppression in the local interconnect, their resistance and delay are still larger than Cu. CNT-based interconnects also have critical process requirements for defects, contacts, and variability in the local stage. Based on the reduced resistance caused by long MFP, CNTs with high purity and high metallic fraction begin to show performance beyond that of Cu in the intermediate interconnects. At the global level, not only the electrical performance surpasses Cu, but also the advantages in reliability are fully exerted, such as high thermal conductivity, large current carrying capacity, and large EM resistance. The integration process for achieving high density and large MFP would be the key solution to enhance the performance of CNT interconnects on the global stage. In TSV applications, the excellent reliability of CNTs is demonstrated again. Nevertheless, the electrical property of CNT TSV is still limited by the integration process. A low-temperature Si-compatible CNT growth process is always necessary for VLSI circuit manufacturing. The emergence of Cu-CNT composites makes up for the deficiency of CNT’s electrical property and achieves both excellent electrical conductivity comparable to Cu and improved reliability. The performance, reliability, and process requirements of CNT-based interconnects at different lengths are briefly summarized in Table 4.

Integration processes and compatibility are always the key challenges. Contact, density, chirality, defects, and CNT growth temperature (process temperature) have always been obstacles that need to be overcome in the development of CNTs. In particular, some progress has been made in reducing the CNT growth temperature, which improves compatibility. Carbon nanotube technology provides opportunities to realize power-consumption-oriented, energy-efficient on-chip interconnect applications. Thus, CNT-based interconnects have the potential to promote 3D integration, heterogeneous integration, and other directions to achieve high-performance, low-cost, high-energy-efficiency architectures.

## Figures and Tables

**Figure 1 micromachines-13-01148-f001:**
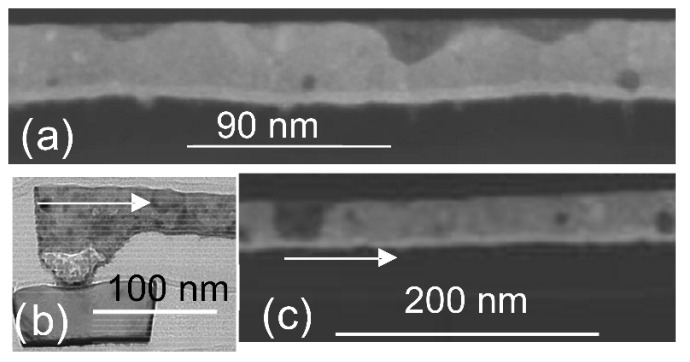
Transmission Electron Microscope (TEM) images of Cu lines and via, including the EM-induced via void and line voids. (**a**) is an unstressed sample of Cu line with Cu fill voids and surface gouges. (**b**) is the via after EM stress with EM-induced voids. (**c**) is the M1 (metal 1) line after stress with EM-induced voids. The arrows show directions of electron flows. Reprinted with permission from ref. [6]. Copyright 2018, IEEE.

**Figure 2 micromachines-13-01148-f002:**
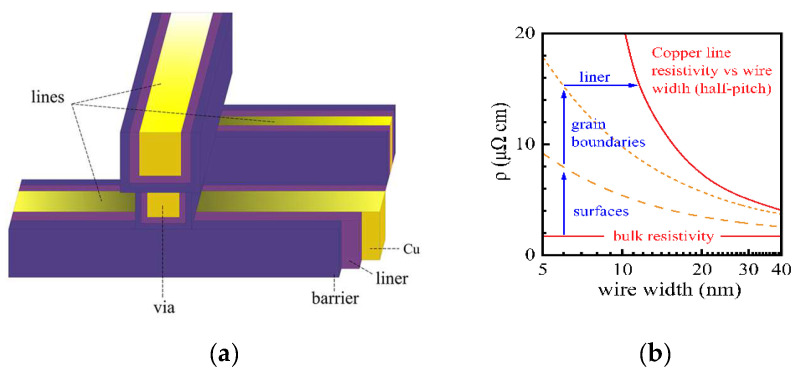
(**a**) Schematic of the interconnect structure. Two layers of horizontal metal lines are connected vertically by via. Barrier and liner are indicated. (**b**) The resistivity of Cu interconnects wire versus wire width. Reprinted with permission from ref. [7]. Copyright 2021, Springer Nature.

**Figure 3 micromachines-13-01148-f003:**
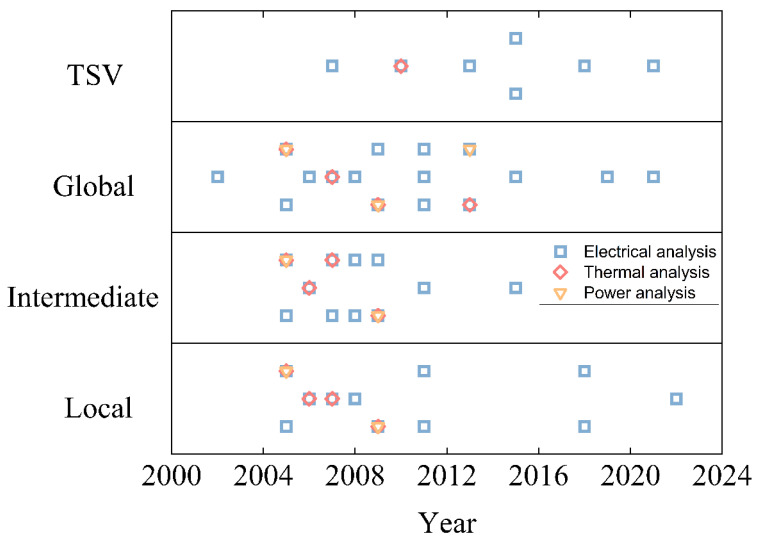
A timetable summarizing recent developments in CNT-based interconnects on local, intermediate, global, and TSV stages with their corresponding electrical, thermal, and power analysis. For TSV level, square patterns represent the related references [27,32,33,34,35,36,37], from left to right respectively. Rhombus patterns represent the related reference [32]. For global level, square patterns represent the related references [20,21,22,23,24,38,39,40,41,42,43,44,45,46,47,48], from left to right respectively. Rhombus patterns represent the related references [23,24,45,49], from left to right respectively. Triangle patterns represent the related references [23,24,44], from left to right respectively. For intermediate level, square patterns represent the related references [21,22,23,24,38,39,40,42,46,50,51], from left to right respectively. Rhombus patterns represent the related references [23,24,39,52], from left to right respectively. Triangle patterns represent the related references [23,24] from left to right respectively. For local level, square patterns represent the related references [21,22,23,24,38,39,41,53,54,55,56], from left to right respectively. Rhombus patterns represent the related references [23,24,39,52], from left to right respectively. Triangle patterns represent the related references [23,24], from left to right respectively.

**Figure 4 micromachines-13-01148-f004:**
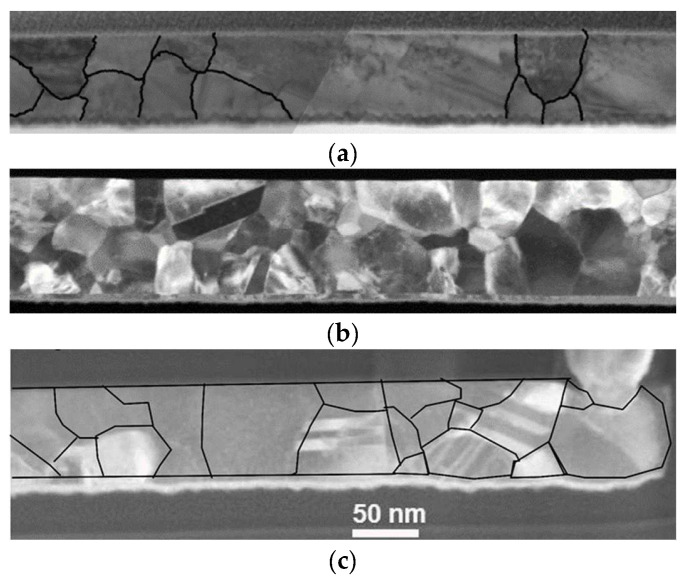
TEM images along Cu (**a**) 80 nm (**b**) 50 nm, and (**c**) 28 nm wide lines. Black lines represent grain boundary locations. Reprinted with permission from ref. [60]. Copyright 2013, Springer Nature.

**Figure 5 micromachines-13-01148-f005:**
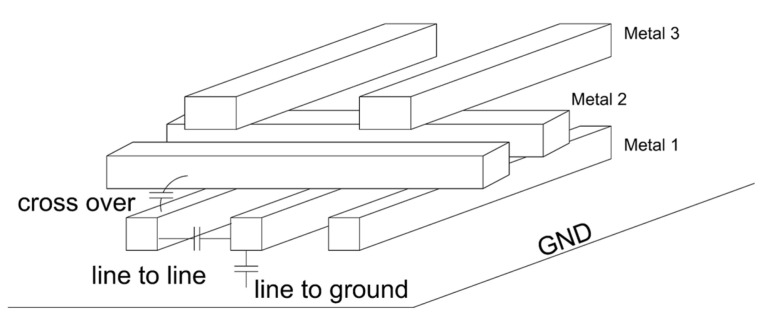
Schematic of interconnect capacitance model. Line to line, line to ground, and cross-over capacitance are shown accordingly.

**Figure 6 micromachines-13-01148-f006:**
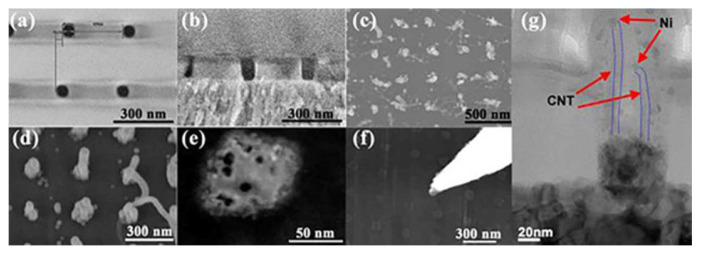
Scanning electron microscope (SEM) images of CNT vias. (**a**) Top-view image after patterning; (**b**) cross-section image after etching; (**c**) image of CNT vias; (**d**) image of CNT vias after dielectric filling; (**e**) top-view of a single CNT after ion milling; (**f**) nanoprobe landing on single CNT and making contact with CNT tips; (**g**) image of a 60 nm CNT via before ion milling, where red arrows indicate Ni catalyst particle at CNT tip and blue lines indicate the CNT sidewalls. Reprinted with permission from ref. [9]. Copyright 2015, IEEE.

**Figure 7 micromachines-13-01148-f007:**
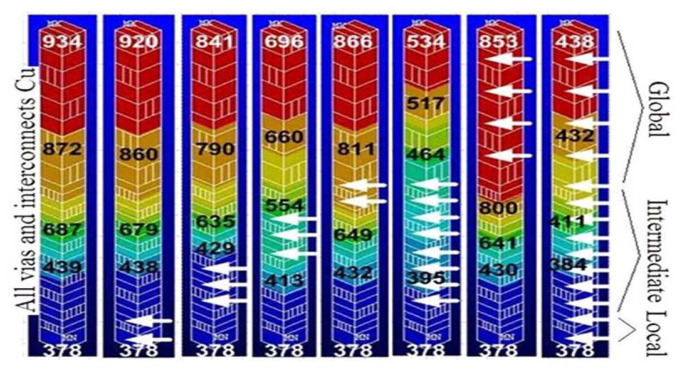
Interconnect temperature map of vias at different levels obtained from 3D finite-element electrothermal simulation at 22 nm node. The parts pointed by the white arrows are composed of CNT bundles, and the other parts are composed of Cu. Reprinted with permission from ref. [24]. Copyright 2009, IEEE.

**Figure 8 micromachines-13-01148-f008:**
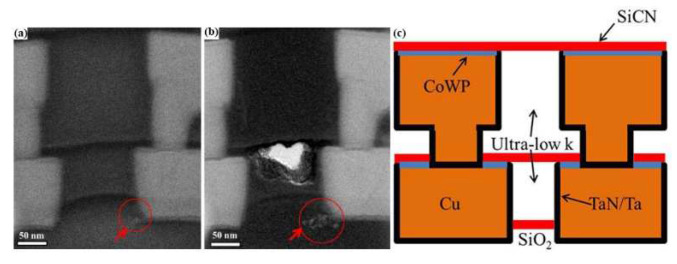
Electron spectroscopic imaging (ESI) analysis of the Cu distribution after the electrical test of (**a**) 20 V/106 min + 25 V/638 min and (**b**) 20 V/106 min + 25 V/872 min. (**c**) is the schematic structure of (**a**,**b**). The structure consists of Cu interconnects in M1 and M2 metal layers, which are encapsulated by a TaN/Ta barrier, SiCN capping layer, CoWP top coating, and insulated by ultralow dielectric permittivity material (porous organosilicate glass). Reprinted with permission from ref. [97]. Copyright 2015, Elsevier.

**Figure 9 micromachines-13-01148-f009:**
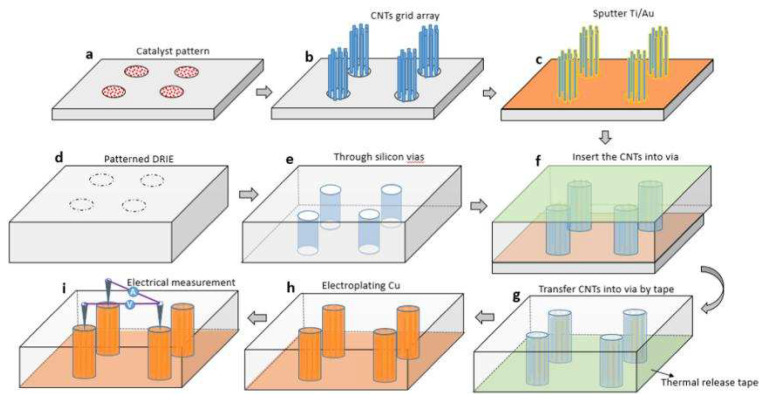
Process flow diagram of Cu-CNT TSV fabrication. (**a**) Patterned catalysts were deposited by E-beam evaporation. (**b**) A CNT grid array was synthesized by using CVD. (**c**) Sputtered 10 nm Ti and 20 nm Au onto the CNT grid array. (**d**,**e**) In parallel to these steps, the target Si wafer/chip with via was prepared by deep reactive ion etching (DRIE). (**f**) Thermal release tape was attached onto the front surface of the target wafer/chip and the CNT grid array were transferred into the via. (**g**) The donor wafer/chip was removed. (**h**) Cu was transferred into the vias by electroplating to form the composite CNT/Cu TSV and the adhesive tape was removed. (**i**) Electrical performance was characterized by the four probe method. Reprinted with permission from ref. [28]. Copyright 2016, IOP Publishing.

**Figure 10 micromachines-13-01148-f010:**
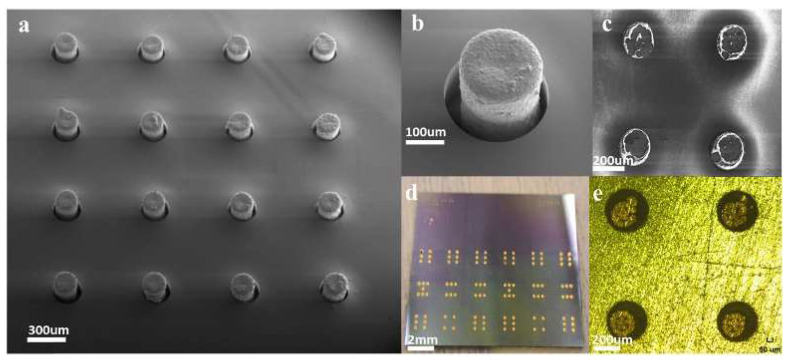
Images of Cu-CNT composite TSVs. (**a**) Cu-CNT composite structures inside the silicon vias; (**b**) the image of a Cu-CNT composite in the via; (**c**) the surface topography of Cu-CNT composite TSVs after polishing; (**d**) a test sample with Cu-CNT TSVs; (**e**) the image showed top surface topography of Cu-CNT composite TSVs after polishing. Reprinted with permission from ref. [28]. Copyright 2016, IOP Publishing.

**Table 1 micromachines-13-01148-t001:** The Development of On-chip Interconnects.

Technology Node	Material	Advantage	Limitation	Industry
14 nm	Cu/W [8,9]	Lower resistivity	Barrier effect	Intel
10/7 nm	Co [5,10]	Barrierless Thin liner	High resistivity	Intel/Global Foundries
5/3 nm	Ru [5,10,11,12]	Barrierless Thin liner	Surface scattering	IMEC
<3 nm	GNR, CNT [13,14,15]	Ballistic transport	Integration/Contact resistance	TSMC

**Table 2 micromachines-13-01148-t002:** Typical values for interconnect size.

Type of Interconnect	Dimensions
Local	<~2 μm [57,58]
Intermediate	2~100 μm [57,58]
Global	>~100 μm [57,58]

**Table 3 micromachines-13-01148-t003:** A summary of properties of SWCNT, MWCNT, Cu-CNT, Cu Co, and Ru.

	SWCNT	MWCNT	Cu-CNT	Cu	Co	Ru
Conductivity (S/cm)	7 × 10^5^ [83,84]	2.7 × 10^5^ [84]	2.3–4.7 × 10^5^ [29,85]	5.8 × 10^5^	1.6 × 10^5^	1.4 × 10^5^
Thermal Conductivity @300k (W/mK)	>3500 [52]	3000 [81]	637 [86]	385	100	117
Electron mean free path @300K (nm)	>1 μm [87]	>30 μm [88]	NA	39	19 [3]	6.7 [3]
Dielectric constant *k* *	graphene oxide-polyimide (*k* = 2) [89]	graphene oxide-polyimide (*k* = 2) [89]	NA	SiCOH (*k* = 2.4~2.55) [58]	SiCOH (*k* = 2.7~3.2) [58]	SiCOH (*k* = 2.4) [11]

* For air gap, *k* = 1 can theoretically be used as long as the process is compatible.

**Table 4 micromachines-13-01148-t004:** A summary of the performance, reliability, and process requirements of CNTs at different length scales by comparing with Cu.

	Local	Intermediate	Global	TSV
Performance	Inferior to Cu (non-ideal)	~30% lower delay	>30% lower delay 8 times less power consumption 7 times larger thermal conductivity	Inferior to Cu (non-ideal)
	Exceeds Cu (ideal)			Exceeds Cu at high frequency (ideal)
				Better thermal performance
Reliability	Exceeds Cu	Exceeds Cu	Exceeds Cu	Exceeds Cu
Process requirements	Defectless CNTs [50], Low variability [53], Low contact resistance [53], Low temperature (Si process compatible) [36]	Defectless CNTs [50], Dense bundle, High purity [50,51], High metallic fraction [22], Low temperature (Si process compatible) [51]	Dense bundle [39], Large electron MFP [39], High metallic fraction [22], Low temperature (Si process compatible) [117]	Dense bundle [118], Large electron MFP, High metallic fraction [32], Cu-CNT composite [28], Low temperature (Si process compatible) [37]

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
