# Peer review of "Recent Progress and Challenges Regarding Carbon Nanotube On-Chip Interconnects"

_micromachines, 2022, doi:10.3390/mi13071148_

Round 1
Reviewer 1 Report
Thank you for submitting your manuscript to Micromachines. Please see the suggested revisions listed below. Good luck.
Abstract
Line 16 Grammar “nanotubes have attracted”
Intro
29 Grammar “have”
47 Grammar “interconnects are”
49 Define IMEC here. Grammar “was to replace”
Figure 1 Please describe Figures 1a, 1b, and 1c in this caption
63 Grammar “seeking a candidate”
64 Tense “ studied”
66 Tense “explored”
67 Tense “expanded”
69 Plural “interconnects”
70 Define TSMC here.
72 Plural “nanotubes are”
73 Grammar “do to their long”
75 Grammar “a great challenge”
Chapter 2
87 Grammar “interconnects are the”
91 Grammar “interconnects have a length greater than”
Figure 3 Please include scale bars on Figure 3a and 3b.
131 Grammar “crosstalk greatly increase”
134 Grammar “nanotubes”
136 Grammar “nanotubes”
138 Grammar “nodes”
162 Grammar “improved compared to”
188 Grammar “CNTs not only improve latency”
199 Grammar “They carry power”
213 Grammar “and an optimized RC delay”
Figure 5 Please consider explaining specifically in the caption what each of the 5a-5g images are showing. Consider describing a-g in the text as well. Explain why you are showing so many images.
Figure 6 Consider installing a scale bar to indicate the size of the modeled samples. For example, what is the distance from bottom to top?
Figure 7 Please label the materials in the photos and highlight the damage in Figure 7b.
Chapter 3
265 Grammar. TSV should be plural, TSVs.
274 Grammar “compared to the horizontal”
275 Grammar “growing CNTs directly in”
277 Grammar “combined structures”
282 Grammar “temperatures”. State the 550 degree temperature units, C, F, or K.
283 Grammar “Experiments show that”
298 Grammar “TSVs have”
300 Grammar “have been investigated”
Figure 9 Increase the clarity of the scale bars. Describe the contents of each of the Figures a-e.
Conclusion
378 Grammar “we reviewed”
380 “CNTs have good”
388-389 Combine to form one sentence. Sentence should not begin with “And”.
391 “chirality”, not “charity”
395-397 Incomplete sentence. Please correct.
Author Response
Please see the attchment. Thank you.

Reviewer 2 Report
In this work, the authors review carbon nanotube-based interconnects from the perspective of different interconnect lengths and through-silicon-via applications. The manuscript is clear and detailed, the conclusions are in accordance with the presented literature, therefore, it can be considered for publication if the following major revision is considered:
- As the authors review the advantages, recent developments, and dilemmas of carbon nanotube-based interconnects, a table or a scheme should be presented to became easier to the reader understand the interest of such interconnects.
- Also, a table with the properties (conductivity, resistivity, thermal conductivity, dielectric constant,….) of these carbon nanotube-based interconnects should be presented. Please add the characteristics of other materials as copper in similar conditions for comparison.
- In the conclusions, the authors state that “Carbon nanotubes exhibit different properties at different length stages.”, please indicate a table or graph demonstrating such conclusion. This will be nice for the readers.
Reviewer 3 Report
The first criticism is that the literature review is not comprehensive enough. It should be improved while clearly mentioning the contribution of the present work at the end of section 1. Please, make sure that your references are cited in an orderly manner. For instance, [18] is not correctly numbered in the caption of Figure 2.
The number of references is sufficient, but the discussion is somewhat simplistic even though CNTs are a hot topic. You should consider taking a look at other similar works and extending the analysis significantly. A timeline comprising the evolution of CNT technology applied in interconnects or even a flowchart would be quite useful to the reader.
[R1] Wei, X., Li, S., Wang, W., Zhang, X., Zhou, W., Xie, S., & Liu, H. (2022). Recent Advances in Structure Separation of Single‐Wall Carbon Nanotubes and Their Application in Optics, Electronics, and Optoelectronics. Advanced Science, 2200054
[R2] Jain, N., Gupta, E., & Kanu, N. J. (2022). Plethora of Carbon nanotubes applications in various fields–A state-of-the-art-review. Smart Science, 10(1), 1-24.
[R3] Kumari B, Pandranki S, Sharma R, Sahoo M. Thermal-Aware Modeling and Analysis of Cu-Mixed CNT Nanocomposite Interconnects. IEEE Transactions on Nanotechnology. 2022 Mar 22;21:163-71.
There is no discussion of fault diagnosis techniques applied to CNT interconnects as well. And what about the thermal analysis and modeling as associated with heat dissipation? Please, elaborate.
Round 2
Reviewer 2 Report
This manuscript has been improved considering the suggestions and, therefore, I can recommend this work for publication